# Dynamic Compressive Stress Relaxation Model of Tomato Fruit Based on Long Short-Term Memory Model

**DOI:** 10.3390/foods13142166

**Published:** 2024-07-09

**Authors:** Mengfei Ru, Qingchun Feng, Na Sun, Yajun Li, Jiahui Sun, Jianxun Li, Chunjiang Zhao

**Affiliations:** 1College of Agricultural Engineering, Shanxi Agricultural University, Jinzhong 030801, China; rumengfei1993@163.com; 2Intelligent Equipment Research Center, Beijing Academy of Agriculture and Forestry Sciences, Beijing 100097, China; sunaswu@email.swu.edu.cn (N.S.); lyj20210043@stu.hunau.edu.cn (Y.L.); sjiahui9@126.com (J.S.); lijianx66@163.com (J.L.); 3Beijing Key Laboratory of Intelligent Equipment Technology for Agriculture, Beijing 100097, China

**Keywords:** tomato, stress relaxation, machine learning, LSTM

## Abstract

Tomatoes are prone to mechanical damage due to improper gripping forces during automated harvest and postharvest processes. To reduce this damage, a dynamic viscoelastic model based on long short-term memory (LSTM) is proposed to fit the dynamic compression stress relaxation characteristics of the individual fruit. Furthermore, the classical stress relaxation models involved, the triple-element Maxwell and Caputo fractional derivative models, are compared with the LSTM model to validate its performance. Meanwhile, the LSTM and classical stress relaxation models are used to predict the stress relaxation characteristics of tomato fruit with different fruit sizes and compression positions. The results for the whole test dataset show that the LSTM model achieves a RMSE of $2.829\times 10^{-5}$ Mpa and a MAPE of 0.228%. It significantly outperforms the Caputo fractional derivative model by demonstrating a substantial enhancement with a 37% decrease in RMSE and a 36% reduction in MAPE. Further analysis of individual tomato fruit reveals the LSTM model’s performance, with the minimum RMSE recorded at the septum position being $3.438\times 10^{-5}$ Mpa, 31% higher than the maximum RMSE at the locule position. Similarly, the lowest MAPE at the septum stands at 0.375%, outperforming the highest MAPE at the locule position by a significant margin of 90%. Moreover, the LSTM model consistently reports the smallest discrepancies between the predicted and observed values compared to classical stress relaxation models. This accuracy suggests that the LSTM model could effectively supplant classical stress relaxation models for predicting stress relaxation changes in individual tomato fruit.

## 1. Introduction

The tomato is one of the most widely cultivated vegetables in the world. Particularly, China emerges as a major producer and consumer of tomatoes. In 2022, tomato yields reached 69.71 million tons, accounting for 34.7% of the total worldwide production [1]. Given that the harvesting, sorting, and packing of tomatoes currently rely heavily on manual labor, and with the increasing cost of labor, there is an urgent need to develop automated machines to replace human efforts. These involved harvest machines, sort machines, pack machines, etc. A common component of these machines is the end effector. For fresh tomato fruit, the mechanical gripping end-effector must ensure reliable holding during the gripping process while also avoiding damage.

The current end-effector for tomato fruit can be divided into three categories as follows: rigid underactuated gripper, soft gripper, and compliance gripper [2]. The rigid underactuated gripper refers to an end-effector whose number of actuators is less than its degrees of freedom, allowing it to passively adapt to the shape of tomatoes [3]. A soft gripper typically operates using a pneumatic drive, which has a long response time and is difficult to control in real time [4]. Compliance gripper can be divided into active and passive compliance. Active compliance achieves grasping action through compliant control methods, while passive compliance adapts to different tomatoes passively through the deformation of materials [5]. Due to the significant differences between tomatoes, passive compliance control faces greater challenges. However, regardless of the gripping method used, all approaches treat tomatoes as rigid bodies and only consider the moment of grasping, thus overlooking the study of viscoelastic properties. While ignoring viscoelasticity, the gripper usually increases the extent of compression to meet the stability of the grasp, which can crush the tomato. On the contrary, it can cause the tomato to slip within the gripper, resulting in abrasions.

The tomato fruit consists of living tissues, such as the locule, pericarp, and septum [6] as the typical parts, with non-uniform viscoelastic properties. Viscoelastic properties, which describe a material’s viscosity and elasticity when subjected to external forces, are particularly important for living tissues. Its typical characteristics are stress relaxation and creep properties. Stress relaxation refers to studying the law of change of stress with time under constant strain conditions [7]. Tomato fruits are subjected to grasping action with typical stress-dynamic relaxation properties [8]. Its magnitude is also related to deformation and contact positions, which also makes the design and control of the end-effector a major challenge.

The traditional viscoelastic mechanical properties of agricultural materials are determined by the standard sampling method. First, experimental data are obtained from standard samples, and then a classical stress relaxation model is fitted to determine the parameters. But its assumption is that agricultural materials are homogeneous and isotropic, such as carrots, cantaloupe, apples, and pears [9,10]. The constitutive law of classical stress relaxation models can be classified into two types as follows: the generalized Maxwell model and the Caputo fractional derivative model [11]. The effectiveness of the generalized Maxwell model in fitting the experimental data, as well as the number of parameters requiring estimation, increases with the number of Maxwell elements [12]. Incorporating more Maxwell elements not only enhances accuracy but also increases model complexity and parameter redundancy, complicating both interpretation and prediction [13,14]. To address the shortcomings, the Caputo fractional derivative model is proposed. The key difference between the two models is that the Caputo model incorporates new viscous elements with fractional derivative characteristics, offering the advantages of high precision and fewer parameters [15].

However, living tissues like tomatoes consist of non-uniform and anisotropic composite materials. This composition makes it difficult to calibrate their relaxation model parameters through standard sampling methods. In recent years, the ability of machine learning models to fit complex relationships has received widespread attention and has been continuously applied to the construction of viscoelastic has received widespread attention and has been continuously applied to the construction of viscoelastic mechanical models for complex materials, which include both biomaterials and non-biomaterials.

The neural network (NN) and generalized KNN, both trained using data generated from stress relaxation tests under different loading conditions, are used to predict the dynamic changes of the force for porcine kidneys [16]. Multilayer perceptron (MLP) is applied to predict the stress relaxation properties of three cultivars of pomegranate at three fruit sizes [17]. Random Forest (RF) is used to predict the displacements of different tissues, such as the liver and breast, under different external forces [18,19]. The radial Basis Function Network (RBFN) is used to predict the relaxation modulus, which is an important parameter in the stress relaxation process [20]. The Recurrent Neural Network (RNN) is used to study the constitutive law of the viscoplastic material [21].

The machine learning model retains the structural anisotropy and non-homogeneous properties of the material and avoids the use of standard sampling methods to calibrate the classical stress relaxation model parameters. The stress relaxation characteristics directly affect the magnitude of gripping force, and its data have typical time series properties. Therefore, we use the LSTM model, which can handle time series data well, for modeling. It can use whole fruit size as the model input parameter, thus making it possible to directly estimate the mechanical properties of various whole fruits.

In view of the need for reliable and safe grasp of tomatoes, this paper proposes a machine learning model based on long short-term memory, where the inputs are time, force, and strain, with the output being stress. It is trained with the dataset from the stress relaxation test to study the dynamic process of stress relaxation characteristics in the whole tomato during grasping. Simultaneously, its accuracy is determined by optimizing the hyperparameters and compared with the classical stress relaxation models (the triple-element Maxwell model and the Caputo fractional derivative model). Different models are also used to predict the stress relaxation characteristics of different single tomatoes, which are compared with each other. The results of this study provide technical support for the development of a tomato fruit end-effector.

## 2. Materials and Methods

### 2.1. Sample Preparation

Provence was selected as the fruit sample because it offered high economic benefits and had a lower hardness at the ripening stage, making it more prone to injury. The samples were selected from red, ripening tomatoes, which were suitable for picking, excluded those with apparent defects and malformed. The samples are shown in Figure 1. The picked tomatoes were kept in a constant temperature and humidity chamber at 5 °C and 80% humidity for more than 10 h to prevent them from ripening further, and the samples were taken out 4 h prior to the test to bring them back to room temperature [22].

### 2.2. Stress Relaxation Data Collection

Stress relaxation tests were accomplished by using a Texture Analyzer (TA.XT.PLUS, SMS, Glasgow, UK) equipped with a column-shaped probe. The contact position between tomatoes and probe was divided into the following two kinds: locule and septum, as shown in Figure 2. Stress relaxation data for the whole tomato was collected by the test along the calyx–stem axis in a vertical direction at 1 mm/s. When the tomato deformation reached 1 mm, the probe was kept for 60 s. The collected data involved the size of the tomato and compress force, and the rest of the parameters are shown in Table 1. The compression stress relaxation tests were repeated 35 times for different contact positions.

### 2.3. Long Short-Term Memory Model

#### 2.3.1. Framework of LSTM and LSTM Unit

The LSTM model is a typical three-layer structure, including an input layer, a hidden layer, and an output layer. Among them, the hidden layer is connected by single or multiple LSTM units. We use two LSTM units [23] to predict relaxation stress, as shown in Figure 3. The model input is a three-dimensional matrix [t,f,ε], and the output [σ] is a one-dimensional vector in this paper. The normalized input is represented by using $\hat{\left[t,f,\varepsilon \right]}$. It can be seen that $\hat{\left[t,f,\varepsilon \right]}$ is transformed into the input layer to form $I^{0}$ and then passed into the LSTM unit, with the following equation:(1)$$I^{0}=w_{I}\hat{\left[t,f,\varepsilon \right]}+b_{I}$$
where $w_{I}$ is the weight of the input layer, $b_{I}$ is the input layer bias column vector, $I^{0}$ is the output of the input layer, t is the sampling time, f is the contact force, and ε is strain, which is the reciprocal of the tomato sizes. Similarly, the output $\hat{\left[\sigma \right]}$ is obtained by using $h^{2}$, $w_{4}$, and $b_{4}$. Here $w_{4}$ is the weight of the output layer, and $b_{4}$ is the bias column vector. Then, the output [σ] is derived by using anti-normalization.

The LSTM unit is mainly composed of four parts, which are three “gates” and historical information states, as shown in Figure 4. The three gates are the forget gate, the input gate, and the output gate. The primary function of the forget gate is to modulate the magnitude of information to be eliminated within the context of historical state, a process executed via a sigmoid activation function. The design objective of the input gate is to regulate the scale of information to be added to the historical information state through a sigmoid function and candidate information state. Information to be added comes from the candidate information state. The output gate aims to determine the scale of the updated hidden layer of memory, operating based on the sigmoid function. The history information state aims to keep and carry information before the current time step.

#### 2.3.2. Forward Propagation of the LSTM Unit

In Figure 3, the input of the LSTM model enters the first LSTM unit, so this paper takes it as an example for illustration. Its input is composed of two parts, namely, the gate input and the history information state, that is, Equation (2):(2)$$in_{lstm}\left(t\right)=\left[in_{gate}\left(t\right),C^{1}\left(t-1\right)\right]$$
where $in_{lstm}\left(t\right)$ is the input of the LSTM unit at time *t*. The gate input $in_{gate}\left(t\right)$ is composed of two parts, which involve $I^{0}\left(t\right)$ and $h^{1}\left(t-1\right)$. $h^{1}\left(t-1\right)$ is the hidden layer state of the first LSTM unit at time t−1. Both $h^{1}\left(t-1\right)$ and $C^{1}\left(t-1\right)$ are concurrently zero at the initial condition. It means that the unit lacks a hidden state and any previously accumulated information at that moment.

First, $in_{gate}\left(t\right)$ and $C^{1}\left(t-1\right)$ need to be submitted to all gates and a historical information state, respectively, and f(t), i(t), o(t), and $\underline{C}\left(t\right)$ are obtained by using Equations (3)–(6). Then, the historical information state $C^{1}\left(t-1\right)$ is updated by using Equations (7) and (8). The last, hidden state $h^{1}\left(t-1\right)$ is refreshed by using Equations (9) and (10). Within the current LSTM framework, the refreshed historical information state $C^{1}\left(t\right)$ is retained for the input of the ensuing time step, as opposed to propagating to the successive unit. The updated historical state information $h^{1}\left(t\right)$ will perpetuate within the current LSTM unit, providing the input for the ensuing time step while simultaneously being relayed to the successive unit.
(3)$$f\left(t\right)=S\left[w_{fI}\cdot I^{0}\left(t\right)+w_{fh}\cdot h^{1}\left(t-1\right)+b_{f}\right]$$
(4)$$i\left(t\right)=S\left[w_{iI}\cdot I^{0}\left(t\right)+w_{ih}\cdot h^{1}\left(t-1\right)+b_{i}\right]$$
(5)$$o\left(t\right)=S\left[w_{oI}\cdot I^{0}\left(t\right)+w_{oh}\cdot h^{1}\left(t-1\right)+b_{o}\right]$$
(6)$$\underline{C}\left(t\right)=T\left[w_{\underline{c}I}\cdot I^{0}\left(t\right)+w_{\underline{c}h}\cdot h^{1}\left(t-1\right)+b_{\underline{c}}\right]$$
(7)$$C^{1}\left(t\right)=f_{t}⊙C^{1}\left(t-1\right)+p\left(t\right)$$
(8)$$p\left(t\right)=i\left(t\right)⊙\underline{C}\left(t\right)$$
(9)$$h^{1}\left(t\right)=o_{t}⊙k\left(t\right)$$
(10)k(t)=T[C(t)]
where f(t), i(t), o(t), and $\underline{C}\left(t\right)$ are the output of forget gate, input gate, output gate, and candidate information state at time *t*, respectively; $C^{1}\left(t\right)$ and $h^{1}\left(t\right)$ are the history information state and hidden state of the first LSTM unit at time *t*, respectively; *S* and *T* are different active functions, which are sigmoid and tanh, as shown in Equations (11) and (12), respectively; $f_{t}⊙C^{1}\left(t-1\right)$ denote the discarded historical information at time *t*, and p(t) is the newly incorporated information at the same temporal juncture; $w_{fI}$ and $w_{fh}$ are the weights of forget gates, which are used to deal with different gate inputs, similarly, the meaning of other weights can be achieved; and $b_{f}$, $b_{i}$, $b_{o}$, and $b_{\underline{c}}$ are the biases of different gates and candidate information states.
(11)$$y=\frac{1}{1+e^{-x}}$$
(12)$$y=\frac{e^{x}-e^{-x}}{e^{x}+e^{-x}}$$

From Equation (11), the range of the sigmoid function is the interval (0, 1). Furthermore, its gradient increases as an input closes to the coordinate origin, indicating a non-symmetrical graph with respect to the origin. From Equation (12), the range of the tanh function is the interval (−1, 1), and its graph is symmetric with respect to the origin. The tanh function can be regarded as a rescaled version of the sigmoid function, and both functions are nonlinear.

#### 2.3.3. Backpropagation of the LSTM Unit

The LSTM model executes the backpropagation mechanism utilizing the BPTT algorithm [24], which is indexed in a reverse temporal direction. Its main purpose is to optimize the training results by continuously changing the weights and biases to bring the loss function down to the desired value. In this paper, the mean square error (MSE) is defined as the loss function, as shown in Equation (13):(13)$$loss=\frac{1}{n}{\sum_{t=1}^{n}\left(\hat{\left[\sigma_{t}\right]}-\sigma_{t}\right)}^{2}$$
where *n* is the time step.

The backpropagation procedure commences with the calculation of gradients for the loss function relative to the trainable parameters. This is essential, given that the antipodal direction of the gradient delineates the most rapid decrease in the loss function. Then, these gradients are transmitted into the network to update the parameters. However, there are many gradients in the LSTM model, for example, the output gate. Its gradients can be divided into three parts, namely, the hidden layer weight, the input layer weight, and the bias, as shown in Equations (14)–(16).
(14)$$\frac{\partial l}{\partial w_{oh}}=\frac{\partial l}{\partial h^{1}\left(t\right)}\cdot \frac{\partial h^{1}\left(t\right)}{\partial o\left(t\right)}\cdot \frac{\partial o\left(t\right)}{\partial m_{o}\left(t\right)}\cdot \frac{\partial m_{o}\left(t\right)}{\partial w_{oh}}$$
(15)$$\frac{\partial l}{\partial w_{oI}}=\frac{\partial l}{\partial h^{1}\left(t\right)}\cdot \frac{\partial h^{1}\left(t\right)}{\partial o\left(t\right)}\cdot \frac{\partial o\left(t\right)}{\partial m_{o}\left(t\right)}\cdot \frac{\partial m_{o}\left(t\right)}{\partial w_{oI}}$$
(16)$$\frac{\partial l}{\partial b_{o}}=\frac{\partial l}{\partial h^{1}\left(t\right)}\cdot \frac{\partial h^{1}\left(t\right)}{\partial o\left(t\right)}\cdot \frac{\partial o\left(t\right)}{\partial m_{o}\left(t\right)}\cdot \frac{\partial m_{o}\left(t\right)}{\partial b_{o}}$$
where $m_{o}\left(t\right)$ is the output of the information preliminarily processed through distinct weights and bias within the forget gate, as shown in Equation (17). So the partial derivatives of $m_{o}\left(t\right)$ with respect to $w_{oh}$, $w_{oI}$, and $b_{o}$ can be obtained by using Equations (18)–(20).
(17)$$m_{o}\left(t\right)=w_{oI}\cdot I^{0}\left(t\right)+w_{oh}\cdot h^{1}\left(t-1\right)+b_{o}$$
(18)$$\frac{\partial m_{o}\left(t\right)}{\partial w_{oh}}={\left[h^{1}\left(t-1\right)\right]}^{T}$$
(19)$$\frac{\partial m_{o}\left(t\right)}{\partial w_{oI}}={\left[I^{0}\left(t\right)\right]}^{T}$$
(20)$$\frac{\partial m_{o}\left(t\right)}{\partial b_{o}}=1$$

Given that o(t) is the result of a sigmoid function with $m_{o}\left(t\right)$ as the independent variable, as shown in Equation (21). The derivative of the sigmoid function is shown in Equation (22). Therefore, its partial derivative with respect to $m_{o}\left(t\right)$ can be obtained, as shown in Equation (23). Similarly, its partial derivative with respect to $h^{1}\left(t\right)$ can be obtained, as shown in Equation (24).
(21)$$o\left(t\right)=S\left[m_{o}\left(t\right)\right]$$
(22)$$\frac{dS\left(Z\right)}{dZ}=S\left(Z\right)\cdot \left[1-S\left(Z\right)\right]$$
(23)$$\frac{\partial o\left(t\right)}{\partial m_{o}\left(t\right)}=S\left[m_{o}\left(t\right)\right]\left[1-S\left(m_{o}\left(t\right)\right)\right]$$
(24)$$\frac{\partial h^{1}\left(t\right)}{\partial o\left(t\right)}=k\left(t\right)$$

By substituting Equations (18), (23) and (24) into Equation (14), we can derive Equation (25). In a similar manner, Equations (26) and (27) are also formulated.
(25)$$\frac{\partial l}{\partial w_{oh}}=\frac{\partial l}{\partial h^{1}\left(t\right)}⊙k\left(t\right)⊙S\left[m_{o}\left(t\right)\right]\left[1-S\left(m_{o}\left(t\right)\right)\right]\cdot {\left[h^{1}\left(t-1\right)\right]}^{T}$$
(26)$$\frac{\partial l}{\partial w_{oI}}=\frac{\partial l}{\partial h^{1}\left(t\right)}⊙k\left(t\right)⊙S\left[m_{o}\left(t\right)\right]\left[1-S\left(m_{o}\left(t\right)\right)\right]\cdot {\left[I^{0}\left(t\right)\right]}^{T}$$
(27)$$\frac{\partial l}{\partial w_{oh}}=\frac{\partial l}{\partial h^{1}\left(t\right)}⊙k\left(t\right)⊙S\left[m_{o}\left(t\right)\right]\left[1-S\left(m_{o}\left(t\right)\right)\right]$$

The update of weights and biases depends on the gradient, as shown in Equation (28).
(28)$$w_{oh}^{new}=w_{oh}^{old}-\frac{\partial l}{\partial w_{oh}}\cdot \alpha$$
where $w_{oh}^{new}$ is the updated weight, $w_{oh}^{old}$ is the weight before being updated, and α is the learning rate. Based on the above equation, the learning rate is used to update the network weights. To achieve better and faster model convergence, it is necessary to reduce the learning rate during training. The learning rate drop period is used to show the mean, where the learning rate is decreased after a certain number of epochs. The learning rate drop factor indicates the extent to which the learning rate decreases each time. Epochs are defined as the number of times the entire dataset is processed by the model. The neurons in the hidden layer are used to construct the LSTM layer. The mini-batch size refers to the amount of data fed into the model at each iteration. In machine learning, data need to be divided into multiple batches and sequentially fed into the model for training, which can improve computational efficiency.

### 2.4. The Classical Stress Relaxation Model

#### 2.4.1. Generalized Maxwell Model

The generalized Maxwell model, which is employed to study the stress relaxation properties of fruit and vegetables, is composed of multiple Maxwell elements arranged in parallel [25]. To balance accuracy and parameter quantity, this paper adopts the triple-element Maxwell model from among the classical stress relaxation models, with its governing equation presented in Equation (29) [26]. The parameters of the triple-element Maxwell model were determined by fitting the experimental data using the least squares method.
(29)$$\sigma \left(x\right)=\left[E_{1}\cdot \exp(-x/T_{1})+E_{2}\cdot \exp(-x/T_{2})+E_{0}\right]\cdot \varepsilon_{0}$$
where σ(x) is the relaxation stress, Mpa; $E_{1}$, $E_{2}$, and $E_{0}$ are the decay elastic modulus and the equilibrium modulus, respectively, Mpa; $T_{1}$ and $T_{2}$ are the relaxation times, s; and $\varepsilon_{0}$ is the initial strain.

#### 2.4.2. Caputo Fractional Derivative Model

The Caputo fractional derivative model has a multivariate structure analogous to the generalized Maxwell model, and its single configuration exhibits a level of fitting accuracy commensurate with that of the triple-element Maxwell model [27]. It has only two parameters less than the five parameters of the triple-element Maxwell model, as shown in Equation (30). The parameters of this model are also determined using the least squares method by fitting the experimental data.
(30)$$\sigma \left(x\right)=\frac{\chi_{\alpha }}{\Gamma (1-\alpha )}\cdot x^{\left(-\alpha \right)}\cdot \varepsilon_{0}$$
where $\chi_{\alpha }$ is the relaxation modulus, Mpa; Γ(⋅) is the Gamma function; α is the fractional order, which ranges from 0 to 1; and $\varepsilon_{0}$ is the initial strain.

### 2.5. Performance Evaluation Criteria

The performances of different models were judged by using different criteria, which were root mean square error (RMSE) and mean absolute percentage error (MAPE). These criteria are shown in Equations (31) and (32).
(31)$$RMSE=\sqrt{\frac{1}{N}\sum_{t=1}^{N}{\left(x_{t}-y_{t}\right)}^{2}}$$
(32)$$MAPE=\frac{1}{N}\sum_{t=1}^{N}\left|\frac{x_{t}-y_{t}}{x_{t}}\right|\times 100$$
where $x_{t}$ is the observed data; $y_{t}$ is the predicted data; and N is the number of predicted data.

Although there are many criteria for representing error, including RMSE, MSE, etc., the RMSE is chosen as a criterion because its unit is consistent with the stress, making it convenient to know. The MAPE is used to measure the relative magnitude of the error.

### 2.6. Model Training

The hyperparameters of the LSTM model are the initial learning rate, learning rate drop period, epoch, the mini-batch size, the learning rate drop factor, and the number of hidden layer neurons. The initial learning rate used for upgrading the weight and bias ranged from 0.001 to 0.009. To better converge the model, it is necessary to reduce the learning rate in the training process. This paper adopts the fixed-epoch reduction method for decreasing the learning rate, within the range of [70, 90]. The epoch is assigned values within the range of [400, 900]. The number of neurons in the hidden layer varies from 40 to 60. The mini-batch size is 128. The learning rate drop factor is 0.9. The model runs on a computer with a 3.0 GHz main frequency.

### 2.7. Data Preprocessing

The data exported from the texture analyzer includes three stages as follows: loading, relaxation, and unloading, and the data format is time and force. However, this paper only studies the stress relaxation stage, so it is necessary to eliminate the data from the loading and unloading stages. In the stress relaxation model, stress is the output; therefore, it is necessary to use Equation (33) to convert force into stress [28].
(33)$$\left\{ \begin{array}{l} E=\frac{3F\left(1-\mu^{2}\right)}{4D^{1.5}}{\left(\frac{1}{R}\right)}^{0.5}\\ \varepsilon =\frac{D}{2R}\\ \sigma =E\varepsilon \end{array}\right.$$
where E is the modulus of elasticity of the tomato, MPa; D is the deformation of the tomato, mm; F is the applied load, N; R is the radius of the tomato, mm; σ is the relaxation stress of the tomato, MPa; and ε is the strain of the tomato.

The dataset is divided into a training set (80%) and a test set (20%), where the test set contains data from large, medium, and small fruit, with different loading positions. The dataset needs to be normalized and anti-normalized. The purpose of normalization is to eliminate the impact of different units and ranges of features on the training results, which include the training speed, stability, generalization ability, and weight updating of the model. All data are transformed into the range of [0, 1] because they do not have negative values during the stress relaxation process. Taking the compression force as an example, the normalization is carried out using the following equation:(34)$$y=(f-f_{\min})\times \frac{y_{\max}-y_{\min}}{f_{\max}-f_{\min}}+y_{\min}$$
where f is the original compression force, $f_{\min}$ is the minimum value of original compression force in the whole dataset, $f_{\max}$ is the maximum value of original compression force in the whole dataset; and y is the value after normalization, where $y_{\max}$ and $y_{\min}$ are 1 and 0, respectively. Then, they are substituted into the above formula to obtain Equation (35), which is used for the other features to repeat this process to complete the normalization.
(35)$$y=\frac{\left(f-f_{\min}\right)}{f_{\max}-f_{\min}}$$

## 3. Results and Discussion

### 3.1. Different Models Performance for the Whole Dataset

#### 3.1.1. LSTM-Based Evaluate Performance

Figure 5 shows the results of different hyperparameters on the train and test sets. Figure 5 illustrates the LSTM model’s robust generalization capability, as evidenced by the minimal disparity between its training and testing outcomes. Specifically, in Figure 5a,b, the model’s MAPE and RMSE demonstrate fluctuating trends with the increment of neurons in the hidden layer. Notably, at 40 neurons, both MAPE and RMSE attain their peak values. In contrast, the testing RMSE forms troughs at $4.941\times 10^{-5}$ Mpa and $4.389\times 10^{-5}$ Mpa when neuron counts are adjusted to 48 and 56, respectively. However, a neuron count of 56 results in a MAPE of 0.666%, which surpasses the lowest observed MAPE of 0.499%. Consequently, this study identifies 48 as the optimal neuron count for the hidden layer.

In Figure 5c,d, the model’s MAPE and RMSE show a saddle and fluctuating trend as the learning rate drop period extends, respectively. The highest RMSE and MAPE values are observed when the learning rate drop period is set to 84 and 72, respectively. In contrast, a learning rate drop period of 90 results in the lowest RMSE and MAPE, at $2.964\times 10^{-5}$ Mpa and 0.231%, respectively. Thus, this study determines that the most effective learning rate drop period for optimizing model performance is 90.

In Figure 5e,f, the model’s RMSE and MAPE show a fluctuating trend with the increment of the initial learning rate. Notably, setting the initial learning rate to 0.004 leads to peak values for MAPE and RMSE. Conversely, for the test set, setting the initial learning rate to 0.008 leads to the lowest observed values of RMSE and MAPE, which are $2.964\times 10^{-5}$ Mpa and 0.231%, respectively.

In Figure 5g,h, both RMSE and MAPE generally show a decreasing trend as the increment of epochs increases, with the notable exception occurring at 550 epochs, where this trend does not hold. For the test set, when the epoch is 550, RMSE and MAPE reach peak values of $8.377\times 10^{-5}$ Mpa and 1.301%, respectively. Conversely, at the 850 epoch, both RMSE and MAPE reach the minimum values of $2.829\times 10^{-5}$ Mpa and 0.228%, respectively. Therefore, the optimal epoch in this study is 850. In summary, the optimal hyperparameter combination for the LSTM model includes a learning rate drop period at 90, an initial learning rate of 0.008, 850 epochs, and 48 neurons.

#### 3.1.2. Parameters of the Classical Stress Relaxation Model

The parameters of the triple-element Maxwell model and Caputo fractional derivative model were obtained by using the nonlinear custom fitting method through a dataset, and the optimal parameters were determined through low RMSE and MAPE. The custom equations were Equations (29) and (30). The specific parameters of the triple-element Maxwell model and the Caputo fractional derivative model are shown in Table 2 and Table 3. Analysis of both tables reveals that the number of parameters in the triple-element Maxwell model, which amounts to five, is 2.5 times that of the Caputo model. This signifies that the former model exhibits greater complexity.

#### 3.1.3. Comparison of Different Models

The performances of different models are shown in Table 4. Compared with other models, both RMSE and MAPE of the LSTM model are the smallest, according to Table 4. Consequently, the LSTM model outperforms all others across the entire dataset, showcasing superior performance. The Maxwell model exhibits a RMSE 1.3 times greater and a MAPE 3.8 times higher than those of the LSTM model, while the Caputo model’s RMSE and MAPE are 1.6 times the LSTM model’s index, respectively. Hence, it can be inferred that the RMSE of the Maxwell model is comparable to that of the Caputo model. However, due to the significant difference in MAPE between them, the Caputo model is considered superior to the Maxwell model. By comparing the approximation complexities of the models, we found that the complexity of the LSTM model is 40 times that of the traditional model. However, the time it takes to predict each time step is less than 1 ms. Therefore, we believe that the real-time performance of this model meets our requirements.

### 3.2. Prediction of Different Models for the Whole Fruit of Single Tomatoes

#### 3.2.1. LSTM Prediction

Figure 6 presents a comparison of the LSTM model’s predicted values against the actual values in the stress relaxation process of tomatoes, taking into account various compression positions and sizes. A significant deviation is observed between the multiple predicted values by the LSTM model and the actual values at t = 0.005 s, except for the predictions at the middle fruit septum. This discrepancy is not exclusive to the moment at t = 0.005 s but extends throughout the time interval of t < 0.3 s.

The phenomenon can be attributed to the following two primary causes: sampling issues in plotting and inherent flaws in the model itself. Initially, the limited number of samples represented in the graph results in significant discrepancies between predicted and actual values being evident only at 0.005 s. Furthermore, in the time interval of t < 0.3 s, the model suffers from a lack of sufficient historical information to rely on, leading to deviations. As time progresses and the model accumulates more historical information, these deviations gradually diminish.

Upon analyzing all the curves depicted in Figure 6, it is noticeable that the maximum relaxation stress observed in Figure 6b,e appears abnormal. A comparison across Figure 6c–f reveals that the maximum relaxation stress at the locule position is consistently lower than at the septum position. However, Figure 6a shows a higher maximum relaxation stress compared to Figure 6b, and their contact force and strain are similarly matched. An examination of Figure 6a,c,e indicates that the maximum relaxation stress initially decreases and then increases as the size of the fruit reduces. In contrast, Figure 6b,d,f exhibit an opposite trend, with the maximum relaxation stress increasing and then decreasing as the fruit size diminishes.

Considering the contact force, the abnormality observed in Figure 6b may stem from the fact that the actual compression location for the tomato is not at the septum, as presumed, but rather at the locule. This misidentification leads to the recorded stress being lower than the actual stress experienced by the fruit. When the compression is applied at the septum, the discrepancies in the maturity levels of the tomatoes depicted in Figure 6a,b could account for the observed phenomena. Specifically, the tomato in Figure 6b might be at a more advanced stage of maturity than the one in Figure 6a, exhibiting a softening effect. This results in its maximum relaxation stress approaching the levels observed at the locule position. Conversely, the phenomenon observed in Figure 6e can be attributed to the tomato’s lower maturity level compared to those in Figure 6a,b, leading to a higher maximum relaxation stress.

In this study, the ripeness of the samples used in the experiment is determined by the color comparison method. Although this method is simple and non-destructive to the tomatoes, it cannot serve as a quantitative feature input to the LSTM prediction model, thus affecting the prediction results of the LSTM. Therefore, in subsequent experiments, we will seek a method to quantitatively represent ripeness that is non-destructive to tomatoes and use it as an input to the LSTM model. This will help reduce the prediction error caused by the difference between the expected and actual ripeness of the tomatoes, thereby further improving the prediction accuracy of the LSTM network.

Table 5 reveals that when the compression position is the locule, the medium fruit exhibits the highest RMSE of $2.631\times 10^{-5}$ Mpa. And the small fruit has the largest MAPE at 0.192%. However, changing compression to septum, the medium fruit demonstrates the smallest RMSE, measured at $3.438\times 10^{-5}$ Mpa and exceeding the former by 31%, and the small fruit demonstrates the lowest MAPE, marked at 0.375% and exceeding the former by 95%. Comparing the results from the entire dataset reveals that the RMSE and MAPE for the locule are lower than those for the entire dataset, while the results for the septum are opposite. Further, in conjunction with Figure 6, it becomes clear that the elevated RMSE and MAPE at the septum position are primarily due to the model’s predictions being consistently higher than the actual values. Additionally, the noticeable fluctuation of the predicted values at the septum position for the large fruit around 0.555 s is another factor contributing to the abnormal RMSE and MAPE.

#### 3.2.2. Comparison of the Predicted Values of Different Models

Figure 7 shows the stress relaxation curves for a single tomato fruit, along with the predictive curves from the LSTM and classic stress relaxation models. In the figure, the predictions made by the triple-element Maxwell model exceed the actual values, with the difference reaching its peak at 0.005 s, and these model predictions enter the plateau phase earlier relative to the actual values. The predictions made by the Caputo fractional derivative model are below the actual values, aligning closely only at 0.005 s and following the same downward trend as the actual values. The predictions of the LSTM model deviate the least from the actual values, with most predictions overlapping the actual values, except at 0.005 s, where the difference is greater.

The observed phenomenon in classical stress relaxation models can be attributed to the fact that, once the parameters of the model are set, the predicted stress magnitude becomes solely dependent on time and strain, neglecting the specific mechanical properties of the sample. Furthermore, according to the formula used in classical stress relaxation models, it is clear that the model is incapable of predicting stress relaxation at different compression locations. This oversight results in a significant discrepancy between the model’s predictions and the actual stress relaxation behavior of the samples, rendering it incapable of accurately depicting the stress relaxation characteristics of different tomatoes.

Therefore, it can be concluded that classical models are not suited for simulating the stress relaxation process of whole tomatoes but are more appropriate for the stress relaxation of standard samples. Among the three models, the difference between observed and predicted values is the smallest for the LSTM model. Therefore, the LSTM model is suitable for predicting the stress relaxation of tomatoes under various conditions, including different compression locations and strains. Currently, our research is centered on tomatoes, and we have not yet studied the adaptability of the LSTM model to other fruits and vegetables with unique configurations, such as apples, pears, berries, and peppers.

## 4. Conclusions

The paper studies the whole fruit of tomatoes, which have non-uniform complex viscoelastic characteristics, and addresses the unclear problem of stress changes when compressed. It proposes a LSTM model to predict the stress relaxation process of whole tomato fruit. Through stress relaxation tests, a dataset of whole fruit compression stress relaxation is established, leading to the following conclusions:

The proposed LSTM model achieved fitting of the stress relaxation characteristics of whole tomato fruits, avoiding the use of standardized sampling methods. Through optimization of the model hyperparameters, the RMSE of the LSTM model reached $2.829\times 10^{-5}$ Mpa, and the MAPE reached 0.228%. Since the highest accuracy among comparison models is the Caputo fractional derivative model, the RMSE of the LSTM model improved by 37%, and the MAPE improved by 36% compared to it. The results indicate that the LSTM model achieved better prediction accuracy for tomatoes at the locule position than at the septum. Specifically, the RMSE for the septum position was recorded at $3.438\times 10^{-5}$ Mpa, which is 31% higher than the maximum RMSE observed at the locule position. Furthermore, the minimum MAPE at the septum was 0.375%, surpassing the maximum MAPE at the locule position by 90%.

## Figures and Tables

**Figure 1 foods-13-02166-f001:**
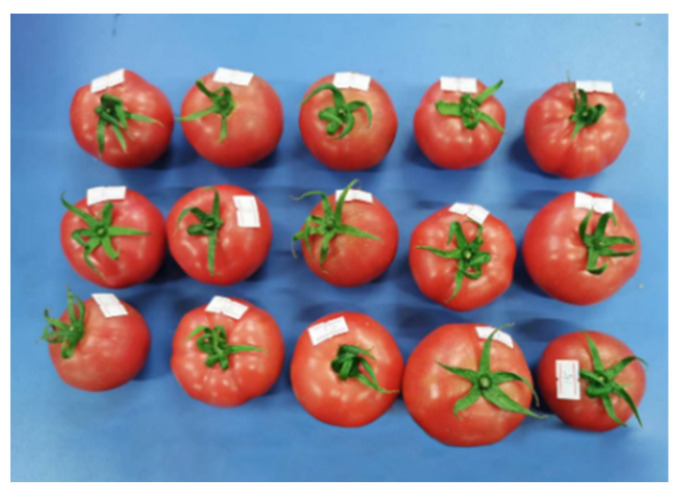
Samples used for relaxation experiments.

**Figure 2 foods-13-02166-f002:**
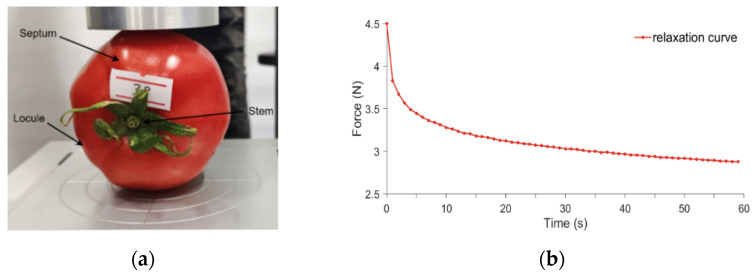
Stress relaxation test. (**a**) is the process of stress relaxation test. (**b**) is its original curve.

**Figure 3 foods-13-02166-f003:**
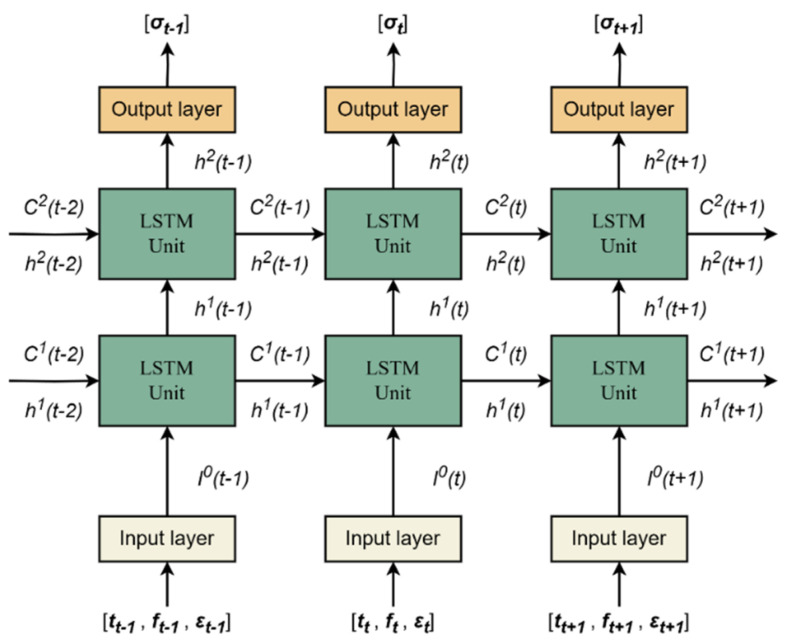
Framework of the LSTM model.

**Figure 4 foods-13-02166-f004:**
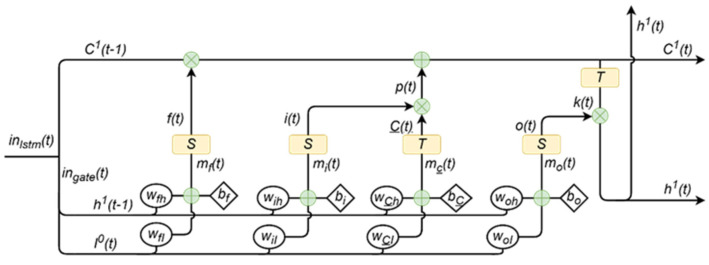
Framework of the LSTM unit.

**Figure 5 foods-13-02166-f005:**
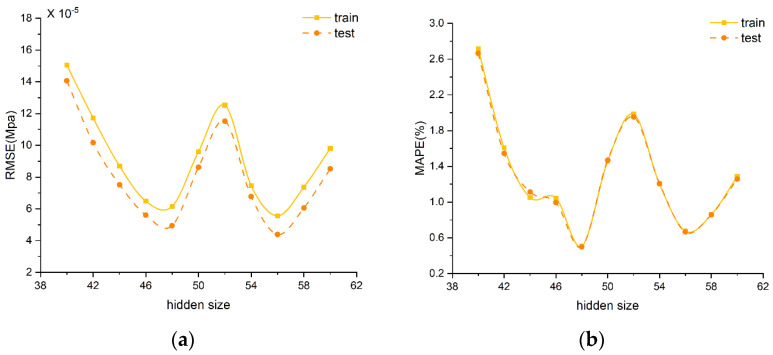
Effect of different hyperparameters on the LSTM network. (**a**,**b**) refer to the hidden size; (**c**,**d**) refer to the learning rate drop period; (**e**,**f**) refer to the initial learning rate; (**g**,**h**) refer to the epoch.

**Figure 6 foods-13-02166-f006:**
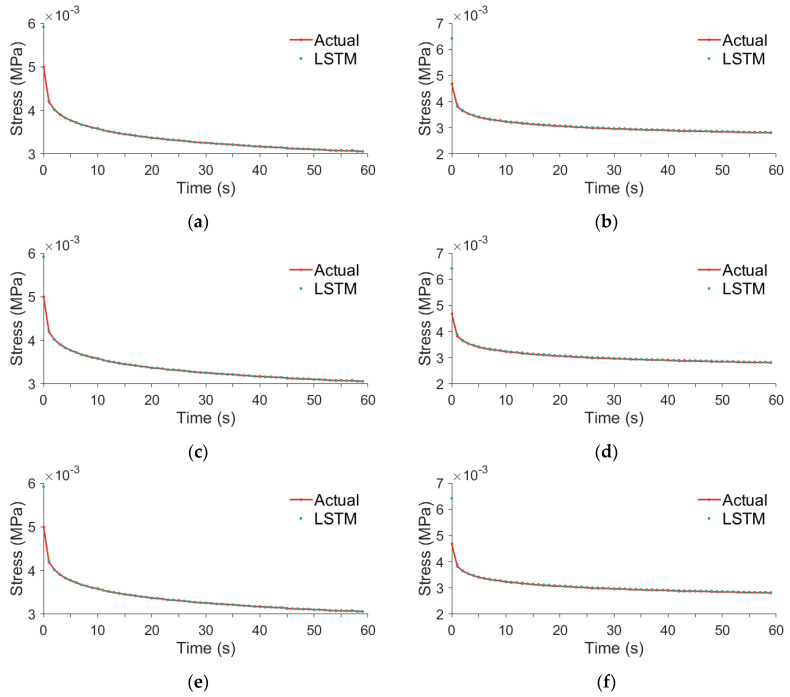
Comparison of predicted and true values for different tomato sizes and compression positions. The compression position of (**a**,**c**) and e is the locule, and the compression position of (**b**,**d**,**f**) is the septum. (**a**,**b**) are the curves of large tomatoes; (**c**,**d**) are the curves of medium tomatoes; and (**e**,**f**) are the curves of small tomatoes.

**Figure 7 foods-13-02166-f007:**
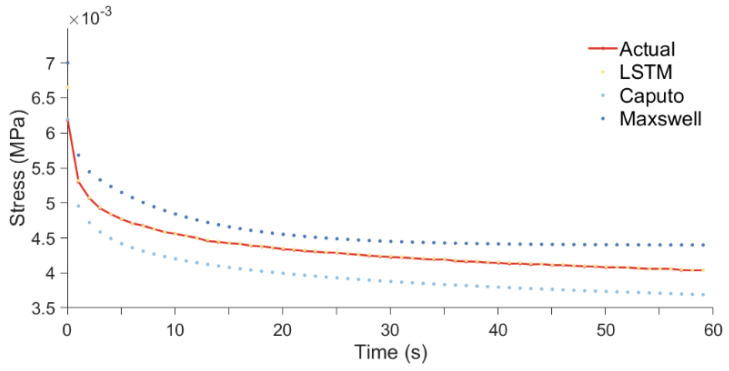
The comparison of predicted values from different models with the actual values.

**Table 1 foods-13-02166-t001:** The parameters of the texture analyzer.

Parameter	Value
Probe diameter	50 mm
Speed of loading	1 mm/s
Speed of loading	20 Hz
Initial force of contact	0.49 N

Note: Probe diameter should be noted so that the probe edge is not contacting the tomato during the test phase to prevent the probe edge from shearing the tomato.

**Table 2 foods-13-02166-t002:** Parameters calculated by the triple-element Maxwell model (mean ± standard deviation).

Materials	*E*_0_ (Mpa)	*E*_1_ (Mpa)	*E*_2_ (Mpa)	*T*_1_ (s)	*T*_2_ (s)
Tomato	0.222 ± 0.0168	0.0681 ± 0.00799	0.0816 ± 0.0144	0.355 ± 0.0517	8.672 ± 0.353

**Table 3 foods-13-02166-t003:** Parameters calculated by the Caputo fractional derivative model (mean ± standard deviation).

Materials	Relaxation Modulus (Mpa)	α
Tomato	0.388 ± 0.025	0.072 ± 0.0012

**Table 4 foods-13-02166-t004:** Performances of machine learning models.

Model	RMSE, (Mpa)	MAPE, (%)
LSTM	$2.829\times 10^{-5}$	0.228
Maxwell	$3.758\times 10^{-5}$	0.867
Caputo	$4.483\times 10^{-5}$	0.355

**Table 5 foods-13-02166-t005:** Prediction results of the LSTM model for different tomatoes.

Data Set		Performance Criterion	
Size	Position	RMSE, (Mpa)	MAPE, (%)
Large	Locule	$1.142\times 10^{-5}$	0.161
	Septum	$7.128\times 10^{-5}$	0.621
Medium	Locule	$2.631\times 10^{-5}$	0.150
	Septum	$3.438\times 10^{-5}$	0.393
Small	Locule	$2.039\times 10^{-5}$	0.192
	Septum	$4.513\times 10^{-5}$	0.375

## Data Availability

The data presented in this study are available on request from the corresponding authors. The data are not publicly available due to privacy or ethical restrictions.

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
