# Peer review of "Dynamic Compressive Stress Relaxation Model of Tomato Fruit Based on Long Short-Term Memory Model"

_foods, 2024, doi:10.3390/foods13142166_

Round 1

Reviewer 1 Report

Comments and Suggestions for Authors

The authors have proposed an exciting solution. I have the following concerns/suggestions,

1.        Figures 5 and 6 are low quality and should be enhanced.

2.        What can be an approximate computational complexity of the proposed model as compared to other similar models on this dataset.

3.        I could not get a clear understanding of adopting LSTM model for this problem domain. The LSTM model mostly addresses time series data is considered computationally expensive.

4.        Tomatoes being variable in size, shape, and ripeness, how the authors manage the data for the model to address this variability.

5.        LSTM models are complex and require a lot of computing power, which can be difficult to use in real-time. How the authors would address it?

Comments on the Quality of English Language

Proofread is required before final submission.

Author Response

Comments 1: Figures 5 and 6 are low quality and should be enhanced.

Response 1: Due to compression, some of the photos in the Figure 5 and 6 became blurry, so we have replaced them. The new Figure 5 is on page 11. And the Figure 6 is on page 13.

Comments 2: What can be an approximate computational complexity of the proposed model as compared to other similar models on this dataset.

Response 2: Traditional fitting and prediction of stress relaxation characteristics are accomplished using the standard sampling method. This method fits Maxwell or Caputo equations based on data obtained from experiments.  After determining the coefficients of the equations, it can predict stress at different time points. Since this method requires fitting different equations, its approximation complexities are O(2.5×104) and O(4×103) respectively. Compared to traditional models, the LSTM model can predict the relaxation stress of tomatoes of different sizes, which also increases the complexity of the model, with a value of O(9×106). However, the model runs on a computer with a 3.0 GHz main frequency, and the time required to predict each time step is less than 1 ms, so we believe it can satisfy the our requirement for real-time performance.

The corresponding revision is on page 9, line 270. It is as follows.

“The LSTM model runs on a computer with a 3.0 GHz main frequency.”

Comments 3: I could not get a clear understanding of adopting LSTM model for this problem domain. The LSTM model mostly addresses time series data is considered computationally expensive.

Response 3: For viscoelastic agricultural materials, stress relaxation is an important characteristic that cannot be ignored, especially during the gripping process by end effectors. The stress relaxation characteristics directly affect the magnitude of the gripping force, making it crucial to use a model to accurately predict the temporal changes in gripping force. The LSTM model performs well in handling time series data; therefore, we have innovatively applied it to the precise prediction of stress relaxation characteristics. We have added the revisions to the second-to-last paragraph of the introduction.

The corresponding revision on page 2, lines 97-99. It is as follows.

“The stress relaxation characteristics directly affect the magnitude of gripping force, and its data has typical time series properties. Therefore, we use the LSTM model, which can handle time series data well, for modeling.”

Comments 4: Tomatoes being variable in size, shape, and ripeness, how the authors manage the data for the model to address this variability.  

Response 4: In reality, tomatoes vary in size, shape, and ripeness, making it evidently inappropriate to use traditional standard sampling methods to predict the relaxation stress of different tomatoes. In the process of harvesting fresh tomatoes, we found a low proportion of malformed fruits. Due to their limited economic value, we excluded them from the sampling range. Therefore, we consider tomatoes to be similar to spherical bodies. This part is located on page 3, line 117 of the manuscript.

This study uses an LSTM model that includes strain as one of the input features, where the strain value is the reciprocal of the tomato diameter.

The corresponding revisions locate on page 4, lines146-147 and page 3, line 103.

This study focuses on tomatoes at the stages of harvesting, sorting, and packaging; therefore, we only examine tomatoes at same ripeness stage. Tomatoes at different ripeness stages may exhibit different colors, but subsequent experiments have shown that tomatoes with the same color can still have different actual ripeness levels. However, in actual production operations, we can only use color, a qualitative method, to determine the ripeness of tomatoes. This method cannot serve as a quantitative feature input to the LSTM prediction model, thus affecting the prediction results of the LSTM. In future work, we will seek a method to quantitatively represent ripeness that is non-destructive to tomatoes and use it as an input to the LSTM model.

The corresponding revision is on page 13, lines 386-393. It is as follows.

“In this study, the ripeness of the samples used in the experiment is determined by the color comparison method. Although this method is simple and non-destructive to the tomatoes, it cannot serve as a quantitative feature input to the LSTM prediction model, thus affecting the prediction results of the LSTM. Therefore, in subsequent experiments, we will seek a method to quantitatively represent ripeness that is non-destructive to tomatoes and use it as an input to the LSTM model. This will help reduce the prediction error caused by the difference between the expected and actual ripeness of the tomatoes, thereby further improving the prediction accuracy of the LSTM network.”

Comments 5: LSTM models are complex and require a lot of computing power, which can be difficult to use in real-time. How the authors would address it?

Response 5: As mentioned in the first issue, the complexity of the model is O(9×10^6). When we deploy the model on a 3.0 GHz processor, the time required for each time step prediction is less than 1 ms. Therefore, we believe that the real-time performance of this method is sufficient to meet the requirements for the end effector to grasp tomatoes.

The corresponding revision is on page 13, lines 346-349. It is as follows.

“By comparing the approximation complexities of the models, we found that the complexity of the LSTM model is 40 times that of the traditional model. However, the time it takes to predict each time step is less than 1 ms. Therefore, we believe that the real-time performance of this model meets our requirements.”

Reviewer 2 Report

Comments and Suggestions for Authors

The manuscript presents an approach to model the dynamic compressive stress relaxation of tomato fruit using Long Short-Term Memory (LSTM) networks. This work contributes to the field of agricultural engineering, particularly in the automation of tomato harvesting and postharvest handling. The results demonstrate that the LSTM model significantly outperforms classical models, which is promising for future applications. However, several issues need to be addressed to improve the clarity, rigor, and completeness of the manuscript.

1.     A clearer explanation of the hyperparameter optimization process would be beneficial. Specifically, detailing what each parameter represents and how they were chosen would enhance the reader's understanding.

2.     The discussion section could be expanded to delve deeper into the implications of the findings. Addressing potential limitations and future research directions, such as the applicability of the model to other fruits or different stages of ripeness, would provide a more comprehensive perspective.

3.     Improving the quality of Figure 5 by adjusting the color and font would enhance readability and clarity for the readers.

4.     Including the standard deviation in many figures is crucial. This addition would suggest the reproducibility of the results and indicate possible errors in the data sets.

Comments on the Quality of English Language

The quality of the English language is good.

Author Response

Comments 1: A clearer explanation of the hyperparameter optimization process would be beneficial. Specifically, detailing what each parameter represents and how they were chosen would enhance the reader's understanding.

Response 1: The hyperparameters of the LSTM model include the learning rate, learning rate decay period, number of epochs, number of hidden neurons, and the mini-batch size. And the chosen of the hyperparameter values is somewhat random and is determined based on experience and preliminary experiments.

The corresponding revision on page 8, lines 222-231. It is as follows.

‘’Based on the above equation, the learning rate is utilized to update the network weights. To achieve better and faster model convergence, it is necessary to reduce the learning rate during training. The learning rate drop period is used to show the mean, where the learning rate is decreased after a certain number of epochs. The learning rate drop factor, which indicates the extent to which the learning rate decreases each time. Epochs defined as the number of times the entire dataset is processed by the model. The neurons in the hidden layer, used to construct the LSTM layer. The mini-batch size refers to the amount of data fed into the model at each iteration. In machine learning, data needs to be divided into multiple batches and sequentially fed into the model for training, which can improve computational efficiency.’’

Comments 2: The discussion section could be expanded to delve deeper into the implications of the findings. Addressing potential limitations and future research directions, such as the applicability of the model to other fruits or different stages of ripeness, would provide a more comprehensive perspective.

Response 2: Stress relaxation is physical property of viscoelastic materials. The physicochemical properties of tomatoes at different ripening stages vary, leading to differences in their relaxation characteristics. Modeling to predict stress relaxation essentially reflects the mechanical properties of tomatoes at different ripening stages. For processes such as tomato harvesting, sorting, and packaging, tomatoes are mainly at the red ripe stage. Therefore, our model is primarily focused on tomatoes at this stage. We add some revisions about the applicability of the model to other fruits.

The corresponding revision on page 14, lines 433-435. The content is as follows.

‘’ Currently, our research is centered on tomatoes, and we have not yet studied the adaptability of the LSTM model to other fruits and vegetables with unique configurations, such as apples, pears, berries, and peppers. ‘’

Comments 3: Improving the quality of Figure 5 by adjusting the color and font would enhance readability and clarity for the readers.

Response 3: Due to compression, some of the photos in the Figure 5 became blurry, so we have replaced them. The new Figure 5 is on page 11.

Comments 4: Including the standard deviation in many figures is crucial. This addition would suggest the reproducibility of the results and indicate possible errors in the data sets.

Response 4: The standard deviation is important for the coefficients of the Maxwell and Caputo equations, as it represents the variation in the coefficients among different individuals. Therefore, in Tables 2 and 3, we listed the average coefficients as the coefficients. The standard deviation does not appear in the machine learning results either, because we only trained the model once. They are on page 12, lines 335-336.
